

# A gene expression study of dorso-ventrally restricted pigment pattern in adult fins of *Neolamprologus meeli*, an African cichlid species

Ehsan Pashay Ahi and Kristina M. Sefc

Institute of Zoology, Universitätsplatz 2, Universität Graz, Graz, Austria

## ABSTRACT

Fish color patterns are among the most diverse phenotypic traits found in the animal kingdom. Understanding the molecular and cellular mechanisms that control in chromatophore distribution and pigmentation underlying this diversity is a major goal in developmental and evolutionary biology, which has predominantly been pursued in the zebrafish model system. Here, we apply results from zebrafish work to study a naturally occurring color pattern phenotype in the fins of an African cichlid species from Lake Tanganyika. The cichlid fish *Neolamprologus meeli* displays a distinct dorsal color pattern, with black and white stripes along the edges of the dorsal fin and of the dorsal half of the caudal fin, corresponding with differences in melanophore density. To elucidate the molecular mechanisms controlling the differences in dorsal and ventral color patterning in the fins, we quantitatively assessed the expression of 15 candidate target genes involved in adult zebrafish pigmentation and stripe formation. For reference gene validation, we screened the expression stability of seven widely expressed genes across the investigated tissue samples and identified *tbp* as appropriate reference. Relative expression levels of the candidate target genes were compared between the dorsal, striped fin regions and the corresponding uniform, grey-colored regions in the anal and ventral caudal fin. Dorso-ventral expression differences, with elevated levels in both white and black stripes, were observed in two genes, the melanosome protein coding gene *pmel* and in *igsf11*, which affects melanophore adhesion, migration and survival. Next, we predicted potential shared upstream regulators of *pmel* and *igsf11*. Testing the expression patterns of six predicted transcriptions factors revealed dorso-ventral expression difference of *irf1* and significant, negative expression correlation of *irf1* with both *pmel* and *igsf11*. Based on these results, we propose *pmel*, *igsf11* and *irf1* as likely components of the genetic mechanism controlling distinct dorso-ventral color patterns in *N. meeli* fins.

Corresponding author
Ehsan Pashay Ahi, ehsan.pashay-ahi@uni-graz.at, epa1@hi.is

## INTRODUCTION

The mechanisms underlying the formation and diversification of integumental colour patterns in vertebrates have always been a fascinating subject of biological research. Much

of the scientific attraction to colour patterns, besides an appreciation of their innate beauty, is due to their enormous diversity and rapid evolution, their crucial roles in mate choice and camouflage, and the comparative easiness with which the colour pattern phenotypes can be observed and scored (*Hoekstra, 2006*; *Mills & Patterson, 2009*). The extensive diversity of vertebrate integumental colour patterns arises from variation in migration, organization and differentiation of pigment cells, which themselves originate from neural crest-derived precursor cells during embryonic development (*Kelsh et al., 2009*; *Bronner & LeDouarin, 2012*). It comes with no surprise that teleost fishes as the largest and most diverse group of vertebrates exhibit the highest complexity and diversity in pigmentation patterns. Fish possess the highest number of pigment classes among vertebrates, which are contained in several types of pigment cells (chromatophores), i.e., melanophores (black pigment), erythrophores and xanthophores (yellow-red pigments), cyanophores (blue pigment) and light reflecting iridophores/leucophores (silvery white) (*Fujii, 2000*; *Lynn Lamoreux et al., 2005*; *Kelsh et al., 2009*). The unrivalled diversity of piscine colours and patterns has been attributed to fundamental genetic changes, such as the whole-genome duplication in the teleost lineage providing additional copies of pigmentation genes, and the retention of duplicated pigmentation genes (lost in other vertebrates) from earlier genome duplications in the vertebrate lineage (*Braasch, Volff & Schartl, 2008*; *Braasch et al., 2009*). Despite the differences, genetic studies in fish and tetrapod models, particularly using zebrafish, mouse and chicken mutants, have revealed conservation in some patterning mechanisms such as genes involved in migration and formation of melanophores (*Lister, Close & Raible, 2001*; *Kelsh, 2004*; *Hoekstra, 2006*; *Mills & Patterson, 2009*; *Kelsh et al., 2009*).

Pigment patterning is also highly influenced by environmental cues and cellular interactions between the chromatophores (*Leclercq, Taylor & Migaud, 2009*). Studies in zebrafish showed that the presence of only one type of chromatophore, and the absence of other types, leads to a uniform distribution of the chromatophore throughout the skin without specific pattern (*Singh & Nüsslein-Volhard, 2015*). The interactions between chromatophores can be complex; for instance, iridophores suppress melanophore survival locally but promote it at a longer distance (*Frohnhöfer et al., 2013*). Among the extensive collection of colour patterns, a frequent motif in teleost fish is the organization of pigment cells along dorso-ventral or anterior-posterior body axes into stripes (*Maan & Sefc, 2013*; *Singh & Nüsslein-Volhard, 2015*). So far, several genes have been shown to be involved in stripe formation (*Singh & Nüsslein-Volhard, 2015*). Intriguingly, stripes can appear predominantly or exclusively in specific body compartments and/or fins, and little is known about the molecular mechanisms that restrict pattern formation to specific regions. Moreover, fin and body stripe formation are controlled by different mechanisms in zebrafish, and while body stripe formation has been extensively studied, much less is known about the mechanisms operating in fins (*Singh & Nüsslein-Volhard, 2015*). However, given that fin pattern variation contributes significantly to the phenotypic diversity of teleost fish, an increased understanding of its molecular background is highly desirable.

In this study, we quantitatively assess the expression of a set of genes involved in adult zebrafish pigmentation and stripe formation (Table 1) in distinctly coloured fin regions of a cichlid fish, *Neolamprologus meeli*, endemic to the Lake Tanganyika in Africa. The study

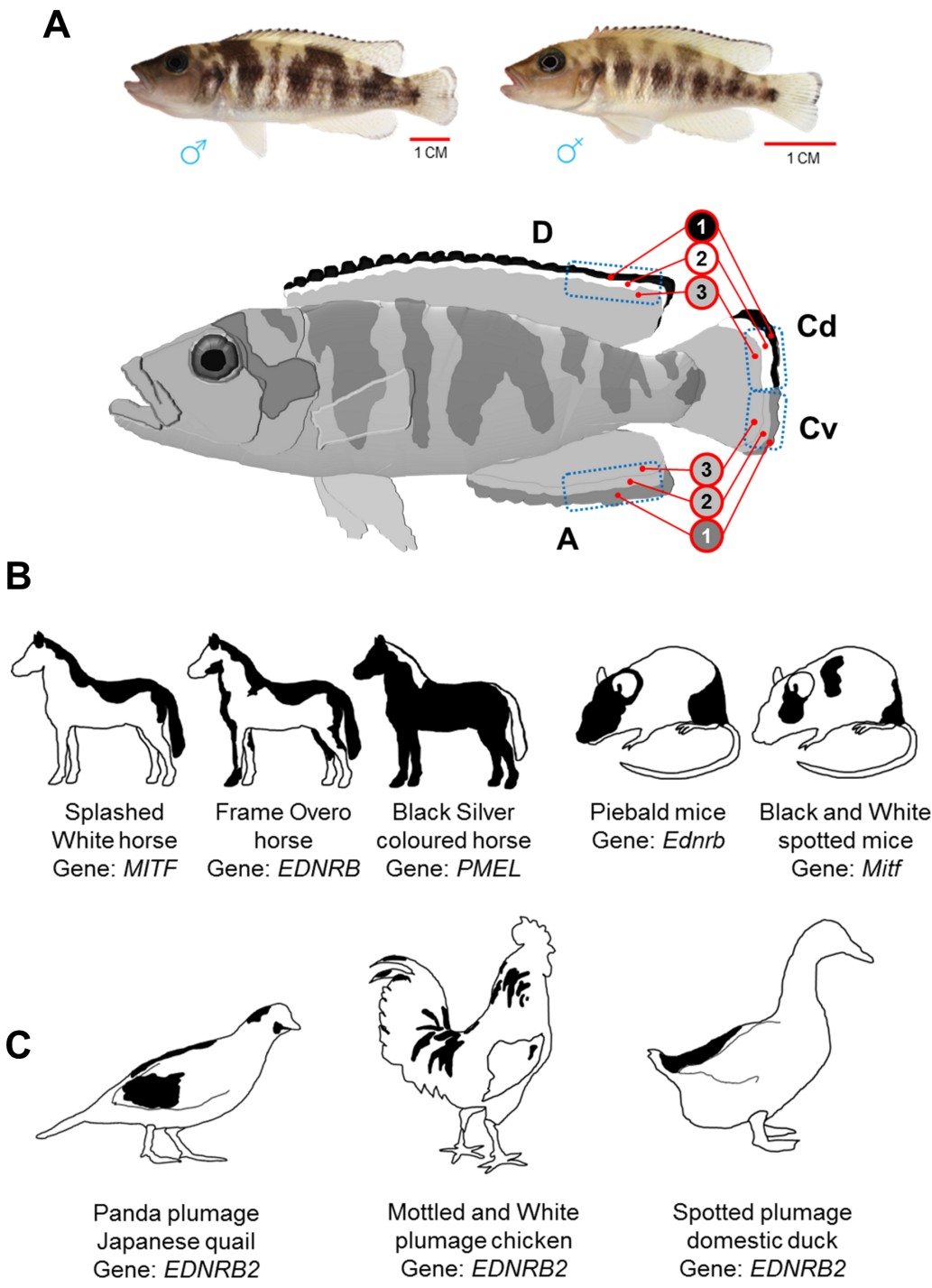

**Figure 1  Dorsally restricted melanin pigmentation patterns in different vertebrate species.** (A) On top, adult male and female Lake Tanganyika cichlid fish, *Neolamprologus meeli*, and below, a schematic drawing displays sharp black–white stripes in the edges of the dorsal fin (D) and the dorsal half of the caudal fin (Cd), whereas the white stripe is missing and the black stripe is replaced by less sharp dark grey stripe in the anal fin (A) and the ventral half of caudal fin (Cv). 
**Figure 1 (…continued)**
The blue dashed squares delineate the fin tissue analysed in the present study; the numbers 1–3 in the red circles distinguish the investigated fin regions based on their coloration and distal to proximal location. (B) Loss-of-function mutations in *Mitf* and *Ednrb* genes cause skin phenotypes with dorsally restricted black pigmentation in horse and mice. A converse colour phenotype has been observed in black Silver coloured horses as a result of a mutation in the *PMEL* gene (only mane and tail are white). (C) Loss-of-function mutations in an avian paralog of the *Ednrb1* gene, *EDNRB2*, cause phenotypes with black–white plumage (dorsal black spots) in Japanese quail, domestic chicken and duck.

**Table 1    Selected candidate target genes and available literature indicating their role in pigmentation and stripe formation in zebrafish.**

| Gene | Developmental formation | | | Adult pigmentation[a] | Stripe formation | References |
|------|-------------|-----------|------------|------------------|------------------|------------|
|      | Melanophore | Iridophore | Xanthophore |                  |                  |            |
| bnc2 | + | + | + | + | + | *Lang et al. (2009)* |
| csf1r | + | ? | + | + | + | *Parichy & Turner (2003)* |
| ece2 | ? | + | ? | + | + | *Krauss et al. (2014)* |
| ednrb1 | + | + | ? | + | + | *Parichy (2006)* |
| fbxw4 | + | + | ? | + | + | *Kawakami et al. (2000)* |
| igsf11 | + | ? | ? | + | + | *Eom et al. (2012)* |
| kita | + | ? | ? | + | + | *Parichy (2006)*, *Mills, Nuckels & Parichy (2007)* and *Dooley et al. (2013a)* |
| kir7.1 | + | − | + | − | + | *Iwashita et al. (2006)* |
| ltk | ? | + | − | + | + | *Fadeev et al. (2015)* and *Lopes et al. (2008)* |
| mitfa | + | ? | − | + | + | *Lister, Close & Raible (2001)* and *Johnson, Nguyen & Lister (2011)* |
| mpv17 | ? | + | ? | + | + | *Krauss et al. (2013)* |
| pmel | + | ? | ? | + | ? | *Schonthaler et al. (2005)* |
| slc24a5 | + | ? | ? | + | − | *Lamason et al. (2005)* |
| slc45a2 | ? | ? | ? | + | ? | *Dooley et al. (2013b)* |
| sox10 | + | ? | ? | ? | + | *Dutton et al. (2001)* and *Elworthy et al. (2003)* |

**Notes.**
[a] A role in adult pigmentation mainly indicates the requirement of gene function for survival of different chromatophore lineages or/and pigment formation in adult zebrafish.

fish displays a contrasting pattern of black and white stripes in its dorsal and caudal fins (see Fig. 1A) which emerges during the juvenile stage and is maintained throughout adulthood. The phenotype resembles dorso-ventrally distinct pigment patterns observed not only in fishes but also in evolutionary distant mammalian and avian species (Figs. 1B and 1C) (*Metallinos, Bowling & Rine, 1998*; *Matsushima et al., 2002*; *Baxter et al., 2004*; *Miwa et al., 2007*; *Hauswirth et al., 2012*; *Andersson et al., 2013*; *Kinoshita et al., 2014*; *Li et al., 2015*).

On the cellular level, the distinct dorso-ventral fin patterning could involve differences in the number of melanophores and iridophores, or in the case of melanophores, changes in size and number of melanosomes (the pigmented organelles in melanophores). We hypothesized that variation in the arrangement and abundance of chromatophores and

in melanosome traits of the adult fish would be reflected in the expression levels of genes involved in stripe formation and adult pigmentation. The identification of such differentially expressed genes will provide a foundation for further understanding of the molecular mechanisms underlying pigment motifs along body axes. Furthermore, mechanisms of stripe formation, particularly those involving iridophore-derived patterning, in the zebrafish body appeared to be different from those in fins (*Singh & Nüsslein-Volhard, 2015*). Hence, our study will shed light on the role of genes, which control the formation of body stripes in adult zebrafish, in the patterning of *N. meeli* fins. In our study, we first characterized the colour pattern phenotype by determining the distribution of melanophores in each fin region, and verifying the presence of iridophores based on the expression of an irodophore-specific marker gene. Next, we investigated the expression patterns of candidate genes in the differently coloured regions of the different fins. Finally, we examined the expression patterns of predicted potential upstream regulators of two differentially expressed target genes. By identifying dorso-ventrally distinct expression of two target genes and one potential shared transcription factor, our work suggests potential components of the mechanism controlling dorsally restricted stripe formation in *N. meeli*.

## METHODS

### Fin sampling and melanophore counting

Four captive bred, adult individuals of *N. meeli*, two males and two females, were used in this study. The size of the fish was between 7–10 cm and their fins looked intact. The fishes did not show sex dependent differences or individual variation in their stripe patterns. Prior to the experiments, the fish were transferred to separate aquaria and fed on identical diets for two weeks. To obtain fin tissue biopsies, the fish were anesthetized in water containing 0.04 gram per litre of MS-222 and parts of their dorsal fin (D), anal fin (A), and dorsal and ventral parts of their caudal fin (Cd and Cv) were cut under a stereomicroscope (the clipped fin areas are specified by blue dashed squares in Fig. 1A). From each of the cut fin tissues, a piece was dissected and immersed in 0.5 mg/ml epinephrine solution (Sigma No. E4375) for 60 min in order to aggregate the melanosomes. The remaining fin tissue was dissected carefully based on its distinct colours (specified with red circles in Fig. 1A), and each piece was immersed in RNAlater (Qiagen) and stored frozen until RNA isolation. Throughout the paper, the individual fin tissues investigated in the study are addressed as "fin regions" and identified by fin type and a number referring to location, such that, for instance, A-1 stands for the distal region of the anal fin, A-2 identifies the middle region and A-3 the proximal region (Fig. 1A). Anatomically equivalent fin regions are grouped into distal, middle and proximal "classes of fin regions" (e.g., the distal class consists of the regions D-1, Cd-1, Cv-1 and A-1; Fig. 1A).

Photographs of the epinephrine treated fin tissues were taken with a camera mounted on a stereomicroscope (eyepiece micrometre x20, Leica). For each fin region, melanophores were counted in four inter-ray fin tissue sections and the area of the inter-ray tissue was measured based on pixel counts. Melanophore density in each fin region was calculated as number of melanophores/mm$^2$. Anaesthesia and fin biopsies were performed under

permit number BMWFW-66.007/0013-WF/V/3b/2016 issued by the Federal Ministry of Science, Research and Economy of Austria.

## RNA isolation and cDNA synthesis

Tissue samples of the individual fin regions (see above) were transferred from RNAlater to tubes containing TRI Reagent (Sigma) and 1.4 mm ceramic spheres, and homogenized by FastPrep-24 Instrument (MP Biomedicals, Santa Ana, CA, USA). RNA was extracted according to manufacturer's Trizol protocol and dissolved in 30 µl RNase-free water. RNA samples were treated with DNase (New England Biolabs) to remove contaminating DNA. RNA concentration was measured by spectrophotometry using a Nanophotometer (IMPLEN GmbH, Munich, Germany). The quality of the RNA samples was evaluated in a R6K ScreenTape System on an Agilent 2200 TapeStation (Agilent Technologies) to ensure that the integrity number (RIN) of all samples was higher than 7. cDNA was prepared from 1,000 ng of RNA using the High Capacity cDNA Reverse Transcription kit (Applied Biosystems), according to the manufacturer's protocol. Negative controls, i.e., reactions without addition of reverse transcriptase (-RT samples), were prepared to confirm the absence of genomic DNA. cDNA was diluted 1:3 times in nuclease-free water for further use in quantitative real-time PCR.

## Gene selection, primer design and real-time qPCR

To validate suitable reference genes for accurate expression analysis, we selected 7 genes expressed in a variety of tissues and frequently used as reference gene candidates in qPCR studies of teleost fishes (Table S1) (*Tang et al., 2007*; *Fernandes et al., 2008*; *Olsvik, Søfteland & Lie, 2008*; *Small et al., 2008*; *Zheng & Sun, 2011*; *Yang et al., 2013*; *Ahi et al., 2013*). In addition, we selected 15 target genes that are known to be involved in adult pigmentation and/or stripe formation in zebrafish (Table 1 and Table S1). Later, we extended our expression analyses to six additional candidate genes which were predicted as potential upstream regulators of two differentially expressed genes in our study (Table S1).

The qPCR primers were designed within sequences conserved across African cichlids, based on recently released transcriptome data from a distantly related species, *Oreochromis niloticus*, and a closely related species from Lake Tanganyika, *Neolamprologus brichardi* (*Brawand et al., 2014*). The sequence alignment was conducted using CLC Genomic Workbench, version 7.5 (CLC Bio, Aarhus, Denmark) and locations overlapping the exon boundaries of the genes were determined based on the Nile Tilapia annotated genome sequences in the Ensembl database (http://www.ensembl.org/Oreochromis_niloticus). The qPCR Primers were designed on exon boundaries of the conserved regions using Primer Express 3.0 software (Applied Biosystems, Foster City, CA, USA) and checked for self-annealing, hetero-dimers and hairpin structures with OligoAnalyzer 3.1 (Integrated DNA Technology) (Table S1).

Real-time PCR was performed in 96 well-PCR plates on an ABI 7500 real-time PCR System (Applied Biosystems) using Maxima SYBR Green/ROX qPCR Master Mix (2X) as recommended by the manufacturer (Thermo Fisher Scientific, St Leon-Rot, Germany). Each biological replicate was run in duplicate together with no-template control (NTC) in
each run for each gene and the experimental set-up per run followed the preferred sample maximization method (*Hellemans et al., 2007*). The qPCR was run with a 2 min hold at 50 °C and a 10 min hot start at 95 °C followed by the amplification step for 40 cycles of 15 sec denaturation at 95 °C and 1 min annealing/extension at 60 °C. A dissociation step (60 °C–95 °C) was performed at the end of the amplification phase to identify a single, specific product for each primer set (Table S1). Primer efficiency values (E) were calculated with the LinRegPCR v11.0 programme (http://LinRegPCR.nl) (*Ramakers et al., 2003*) analysing the background-corrected fluorescence data from the exponential phase of PCR amplification for each primer-pair and those with E less than 0.9 were discarded and new primers designed (Table S1).

## Data analysis

Three different ranking algorithms, BestKeeper (*Pfaffl et al., 2004*), NormFinder (*Andersen, Jensen & Ørntoft, 2004*) and geNorm (*Vandesompele et al., 2002*), were employed to identify the most stably expressed reference genes The standard deviation (SD) based on Cq values of the fin regions was calculated by BestKeeper to determine the expression variation for each reference gene. In addition, BestKeeper determines the stability of reference genes based on correlation to other candidates through calculation of BestKeeper index ($r$). GeNorm measures mean pairwise variation between each gene and other candidates, the expression stability or $M$ value, and it excludes the gene with the highest $M$ value (least stability) from subsequent analysis in a stepwise manner. NormFinder identifies the most stable genes (lowest expression stability values) based on analysis of the sample subgroups and estimation of inter- and intra-group variation in expression levels (*Ahi et al., 2013*; *Pashay Ahi et al., 2016*).

The Cq values of the best ranked reference genes was used as Cq $_{reference}$ in the $\Delta$Cq calculations. For the analysis of the qPCR data, the difference between Cq values ($\Delta$Cq) of the target genes and the selected reference gene was calculated for each target gene; $\Delta$Cq $_{target}$ = Cq $_{target}$ − Cq $_{reference}$. All samples were then normalized to the $\Delta$Cq value of a calibrator sample to obtain a $\Delta\Delta$Cq value ($\Delta$Cq$_{target}$ − $\Delta$Cq $_{calibrator}$). For comparisons of gene expression involving all three classes of fin regions, one biological replicate of D-3 was arbitrarily chosen as calibrator sample. For comparisons restricted to anatomically equivalent fin regions, one biological replicate of D-1, D-2 and D-3 served as calibrator for proximal, middle and distal fin region samples, respectively. Relative expression quantities (RQ) were calculated based on the expression level of the calibrator sample ($E^{-\Delta\Delta Cq}$) (*Pfaffl, 2001*). A two-way ANOVA followed by post hoc Tukey's HSD test was implemented for each target gene to compare RQ values among fins (averaged across regions) and fin regions. To assess similarities in the expression patterns of the target genes, Pearson correlation coefficients ($r$) were calculated in R (http://www.r-project.org).

To identify the potential upstream regulators, we performed motif enrichment analysis on 1 kb promoter sequences of two differentially expressed genes, based on the annotated genome of the Nile tilapia (*Flicek et al., 2013*) using three programs: MEME (*Bailey et al., 2009*), SCOPE (*Carlson et al., 2007*) and XXmotif (*Luehr, Hartmann & Söding, 2012*). We retained the enriched motifs that were present in both promoters and screened for potential

transcription factor (TF) binding sites using STAMP (*Mahony & Benos, 2007*), with the motif position weight matrices (PWMs) retrieved from the TRANSFAC database (*Matys et al., 2003*).

## RESULTS

### Characterization of melanophore distribution in the fin regions

Comparisons of melanophore density and gene expression patterns were conducted between anatomically comparable regions of the dorsal, caudal and anal fins; i.e., either among the total areas cut from each fin (e.g. D-1 + D-2 + D-3 for the dorsal fin), or separately within each of the corresponding fin region classes; the distal fin regions (D-1, Cd-1, Cv-1 and A-1); the middle fin regions (D-2, Cd-2, Cv-2 and A-2); and the proximal fin regions (D-3, Cd-3, Cv-3 and A-3). Expression comparisons between different classes of fin regions were avoided because given the different histological properties of distinct anatomical fin regions along proximal-distal axis, gene expression differences between non-analogous regions could arise for various reasons not associated with colour patterning.

Melanophore density was significantly higher in the black distal regions of the dorsal fin and dorsal part of the caudal fin (D-1 and Cd-1) than in their dark grey ventral counterparts, A-1 and Cv-1 (Fig. 2C). The white dorsal middle regions (D-2 and Cd-2) contained almost no melanophores, whereas melanophore densities were intermediate in their grey ventral counterparts, A-2 and Cv-2, and finally the most proximal regions in all the fins had almost similar numbers of melanophores (Fig. 2C). Melanophore density clearly corresponded with the impression of darkness/lightness of the investigated fin regions. Additionally, the white colour of D-2 and Cd-2 regions appeared to be the result of both melanophore absence and an accumulation of white reflecting iridophores (Fig. 2B). Interestingly, total melanophore numbers summed across fin regions did not differ between fins (Fig. 2D). This indicates that the different fin patterns result from variation in the distribution and perhaps also pigmentation of a constant number of melanophores. In other words, aggregation of melanophores in the black regions and their absence in white regions, determines the dorsal stripes, whereas in their ventral counterpart regions the same number of melanophores is distributed more evenly.

### Validation of reference genes

qPCR-based gene expression analyses depend on comparisons with stably expressed reference genes (*Kubista et al., 2006*), which have to be validated for the species and the specific experimental conditions in each study (*Ahi et al., 2013*). To this aim, we tested the expression of seven reference gene candidates on the cDNA generated from each of the 12 fin regions. The expression levels of the reference gene candidates varied from *actb1*, with the highest expression (lowest Cq) (Fig. S1), to *ef1a* and *hprt1* with the lowest expressions (highest Cq). Next, the genes were ranked based on three algorithms, i.e., BestKeeper, geNorm and NormFinder, and standard deviation (SD) as described in *Ahi et al. (2013)* (Table 2). Among the reference genes, *tbp* ranked first by geNorm and NormFiner and second by BestKeeper analyses (Table 2). Hence, the data indicated high expression stability

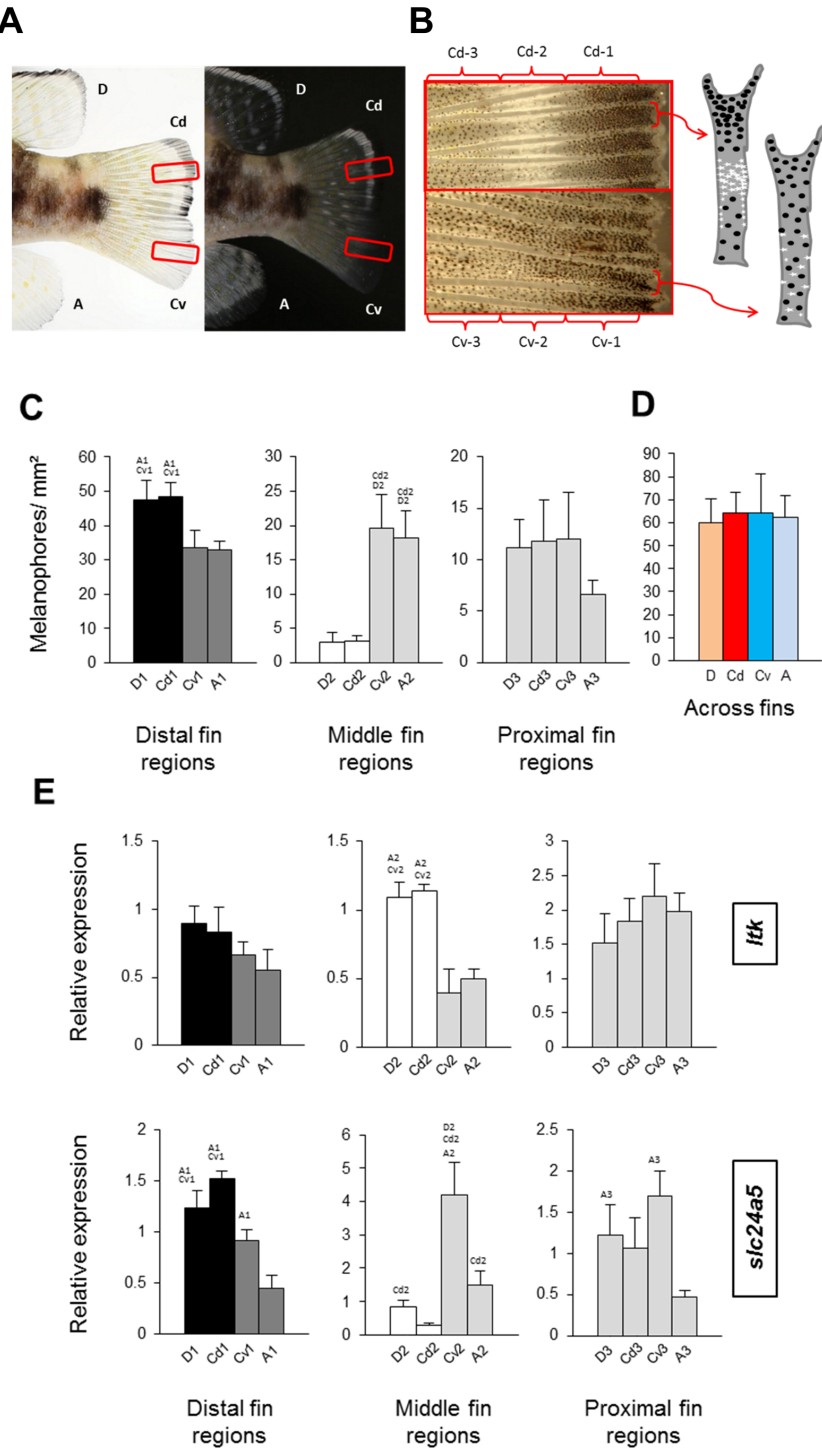

**Figure 2** **Distribution of melanophores and expression of two chromatophore marker genes in fins of**
***N. meeli.*** Fins coded as in Fig. 1A. (A) Photographs of the fins of *N. meeli* taken on white and black background. Insets (red squares) in the caudal fin delineate the magnified area shown in B. (B) The magnification of the caudal fin shows the sequence of black, white and grey coloration in the dorsal part of the fin (fin regions Cd-1, Cd-2, Cd-3, respectively), whereas 

**Figure 2 (…continued)**
colour contrasts are lower in the ventral part of the fin (fin regions Cv-1, Cv-2 and Cv-3, respectively). Analogous patterns were found in the dorsal and anal fins. The chromatophore distribution in the dorsal and ventral parts of the caudal fin is depicted in the schematic drawings of two inter-ray regions (white and black dots represent iridophores and melanophores, respectively). (C) Melanophore densities in the distinct fin regions, grouped by anatomical equivalence into the distal fin regions (D-1, Cd-1, Cv-1 and A-1), the middle fin regions (D-2, Cd-2, Cv-2 and A-2) and the proximal fin regions (D-3, Cd-3, Cv-3 and A-3). Pairwise differences between individual fin regions are indicated by the letter codes of fin regions with significantly lower melanophore density ($P < 0.01$) above bars. (D) Melanophore density summed across fin regions in the D, Cd, Cv and A fins. There were no significant differences between fins. (E) The expression levels of *ltk* and *slc24a5* in the individual fin regions. Statistical comparisons were conducted among anatomically comparable regions, and pairwise differences between fin regions are indicated by the letter codes of fin regions with significantly lower expression ($P < 0.01$) above bars. In (C–E), error bars represent standard deviations calculated from four biological replicates and the comparisons between groups were done by using Tukey-HSD.

**Table 2  Ranking and statistical analyses of candidate reference genes using BestKeeper, geNorm and NormFinder.**

| BestKeeper | | | | geNorm | | NormFinder | |
|---|---|---|---|---|---|---|---|
| Ranking | *r* | Ranking | SD | Ranking | M | Ranking | SV |
| *rps18* | 0.992 | *actb1* | 1.027 | *tbp* | 1.006 | *tbp* | 0.069 |
| *tbp* | 0.987 | *tbp* | 1.099 | *rps18* | 1.023 | *rps18* | 0.113 |
| *rps11* | 0.983 | *ef1a* | 1.160 | *rps11* | 1.030 | *actb1* | 0.141 |
| *actb1* | 0.983 | *rps11* | 1.217 | *actb1* | 1.035 | *rps11* | 0.190 |
| *ef1a* | 0.777 | *rps18* | 1.297 | *ef1a* | 1.470 | *ef1a* | 0.622 |
| *hprt1* | 0.718 | *gapdh* | 1.761 | *hprt1* | 2.285 | *hprt1* | 1.191 |
| *gapdh* | 0.493 | *hprt1* | 2.386 | *gapdh* | 2.352 | *gapdh* | 1.252 |

**Notes.**
Abbreviations: SD, Standard deviation; *r*, Pearson product-moment correlation coefficient; SV, stability value; M, M value of stability.

of *tbp* and suggested it as a suitable normalization factor to accurately quantify differences in gene expression between the fin regions.

## Expression analyses of candidate genes

To assist the cellular characterization of the fin patterns, we investigated the expression levels of an iridophore lineage specific marker *ltk* (*Lopes et al., 2008*) and of the melanosome marker *slc24a5* (*Lamason et al., 2005*) in the individual fin regions. The expression level of *ltk* was significantly higher in the white-coloured regions of the dorsal and caudal fins (D-2 and Cd-2) than in the corresponding ventral regions (A-2 and Cv-2), confirming that iridophores are accumulated in the white stripes. In contrast, expression of *ltk* in the distal and in the proximal fin regions was homogeneous across fins (Fig. 2E) (Table S3). The expression level of *slc24a5* varied significantly among fins for each of the three fin region classes (Fig. 2E). Moreover, *slc24a5* expression was significantly correlated with melanophore density across all fin regions ($r = 0.89$, $p < 0.0001$), although the increase of *slc24a5* expression levels with melanophore number was stronger in the dorsal (D, Cd) than in the ventral (A, Cv) fin tissues (Fig. S2). Variation in the association between melanophore numbers and *slc24a5* expression levels was also observed among fin regions,

as the correlation was significant in the distal and the middle regions ($r = 0.73$, $p = 0.001$; $r = 0.76$, $p = 0.0007$, respectively) but not in the proximal regions of the fins ($r = 0.22$, $p = 0.42$). This suggests that variation in melanosome densities within the melanophores may also contribute to fin color contrasts.

Since the total number of melanophores, summed across regions, was constant in all fins, we next determined the average expression levels of each of the candidate target genes across the three regions of each fin and compared them among fins in order to identify spatial (particularly dorso-ventral) differentiation in gene expression. The selected candidate genes are known to be involved in stripe formation and/or adult pigmentation in zebrafish (see the details in Table 1). One of the genes, *kir7.1* (*kcnj13*), had very low expression levels (Cq > 35) and therefore was discarded from rest of the analysis. This, however, suggests that *kir7.1* expression is not required for chromatophores in fin regions of adult *N. meeli*, which is in agreement with findings in zebrafish, where the function of *kir7.1* was not necessary for pigment cell survival in adults (*Iwashita et al., 2006*). Seven genes, *igsf11*, *ltk*, *mitfa*, *mpv17*, *pmel*, *slc24a5* and *slc45a2*, showed differential expression between the fins (Fig. 3A). The expression level of the melanosome formation gene *slc24a5* was significantly lower in the anal fin than in the other fins. Given that melanophore counts did not differ across fins (Fig. 2D), this indicates a reduced melanosome number in the melanophores of the anal fin. Furthermore, owing to the dorsal white stripes, expression levels of the iridophore marker *ltk* were significantly higher in the dorsal than in the ventral fin tissues. Dorso-ventral differences were also observed in the expression levels of *igsf11*, *mitfa*, *pmel* and *slc45a2* (D, Cd > A, Cv), whereas *mpv17* showed differential expression along the posterior-anterior axis (D, A > Cd, Cv).

Next, we were interested in those genes which showed overall dorso-ventral expression differences across the fins, and compared their expression within the distal, middle and proximal regions separately. Most interestingly, two genes, *igsf11* and *pmel*, displayed higher expression in the black and in the white regions (D-1, Cd-1, and D-2, Cd-2) than in the corresponding ventral regions (Fig. 3B) (Table S3). Importantly, the elevated expression of these genes in the dark stripe was not simply due to the higher density of melanophores in these regions, as the differences persisted after correcting for the number of melanophores (last row in Fig. 3B). Elevated expression in both black and white stripes suggests that *igsf11* and *pmel* have similar expression and potential function(s) in both iridophores and melanophores, which might emanate from their shared neural crest cell origin (*Curran et al., 2010*).

In contrast, the elevated expression levels of two further genes, *mitfa* and *slc45a2*, in the black stripe regions compared to the corresponding ventral regions (first row in Fig. 3B) levelled out after correction for differences in melanophore density among fin regions (last row in Fig. 3B). As discussed before, *ltk* expression was elevated only in the white stripes (Fig. 2E), consistent with its expression in iridophores.

Finally, we extended our study by predicting potential TF biding sites in the upstream promoter sequences of *igsf11* and *pmel* in Nile tilapia, an African cichlid with high quality annotated genome. Using different motif enrichment tools, we identified tens of motifs enriched in the promoter sequences of both genes. After parsing the motifs against the

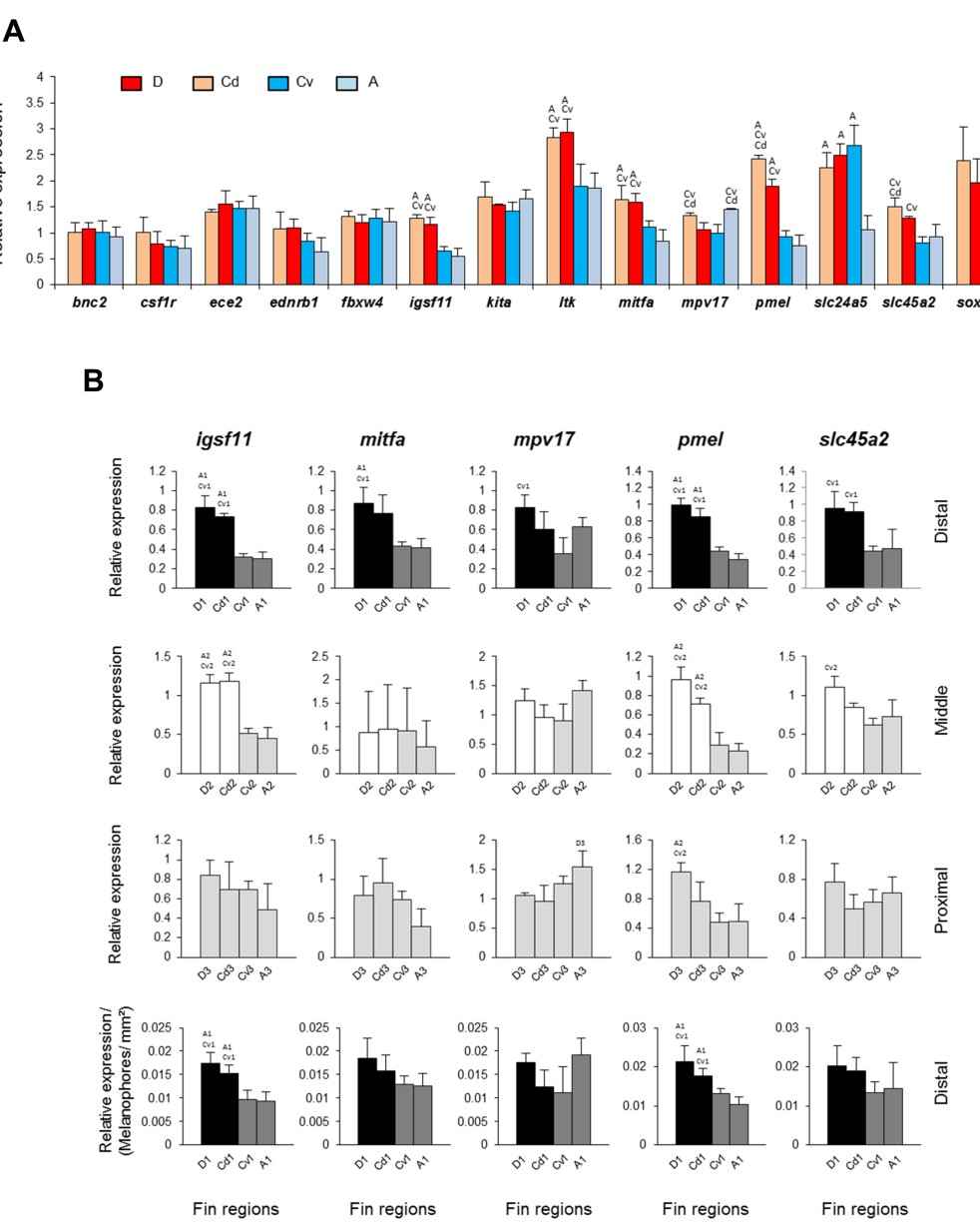

**Figure 3** **Expression differences of target genes among fin samples of *N. meeli*.** Fins and fin regions coded as in Fig. 1A. (A) The overall expression levels per fin, averaged across fin regions, of fourteen candidate target genes. (B) The expression levels of five selected genes, *igsf11*, *mitfa*, *mpv17*, *pmel* and *slc45a2*, compared among anatomically equivalent fin regions. The last row represents expression levels in the distal fin regions corrected for melanophore numbers. Pairwise differences between fin regions are indicated by the letter codes of fin regions with significantly lower expression ($P < 0.01$) above bars. Error bars represent standard deviations calculated from four biological replicates and the comparisons between groups were done by using Tukey-HSD.

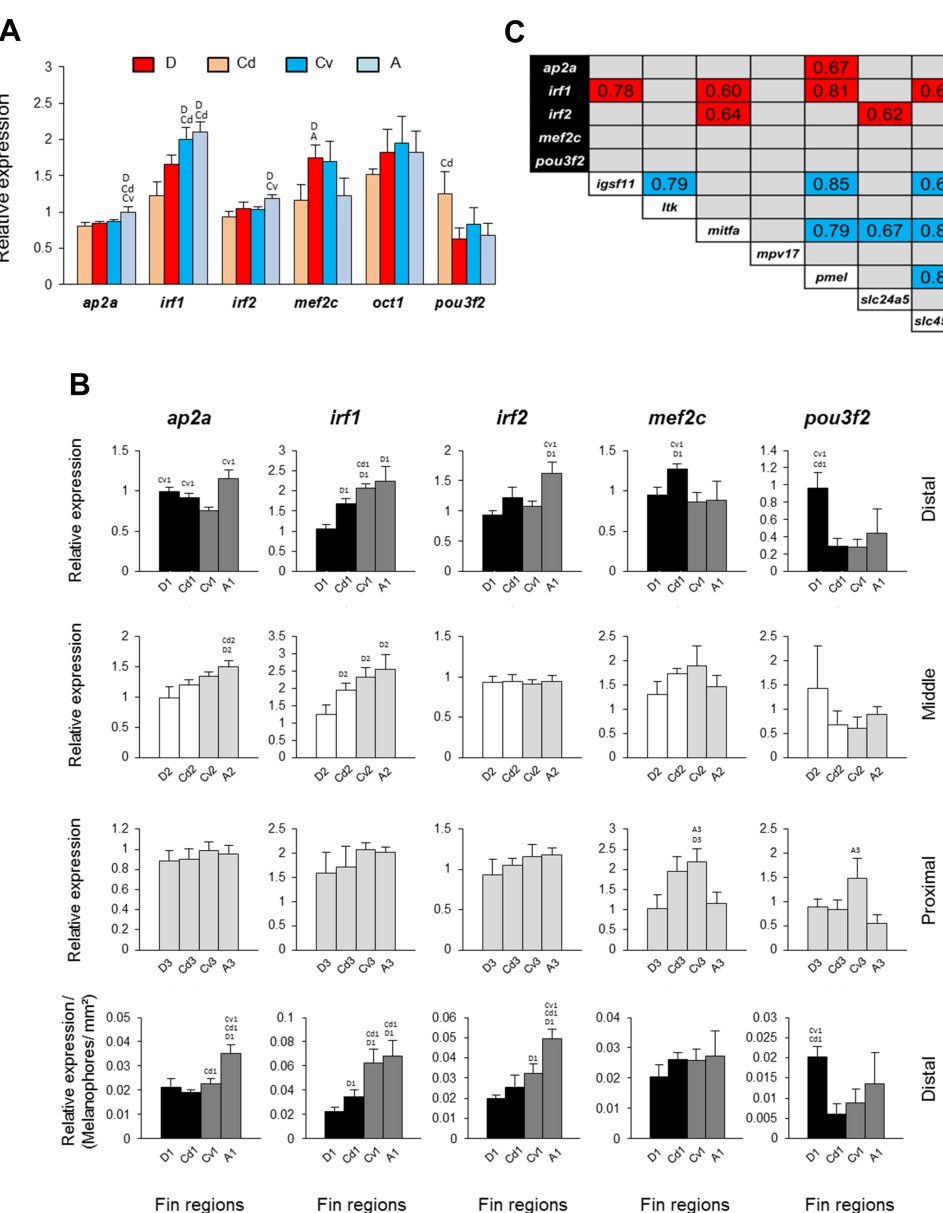

**Figure 4** **Expression differences and expression correlations of predicted transcription factors in fin samples of** ***N. meeli.*** Fins and fin regions coded as in Fig. 1A. (A) The overall expression levels per fin, averaged across fin regions, of six transcription factors. (B) The expression levels of five selected transcription factors, *ap2a*, *irf1*, *irf2*, *mef2c* and *pou3f2*, compared among anatomically equivalent fin regions. The last row represents expression levels in the distal fin regions corrected for the melanophore numbers. Pairwise differences between fin regions are indicated by the letter codes of fin regions with significantly lower expression ($P < 0.01$) above bars. Error bars represent standard deviations calculated from four biological replicates and the comparisons between groups were done by using Tukey-HSD. (C) Significant expression correlations between seven target genes and five transcription factors examined in this study. Pearson correlation coefficients ($r$) were calculated to assess pairwise expression similarities between genes. Blue and red shadings mark positive and negative expression correlation coefficients, respectively, which were significant with $P < 0.01$ (df = 11) in 2-tailed tests.

known TF binding sites in vertebrates, we compiled lists of top potential TFs binding to each motif (Table S2). We then analysed the expression of six of the TFs, which were predicted by all of the three motif enrichment tools and implicated in pigmentation processes (*Barrallo-Gimeno et al., 2004*; *Cook & Sturm, 2008*; *Li et al., 2009*; *Agarwal et al., 2011*; *Besch & Berking, 2014*; *Natarajan et al., 2014*) (Fig. 4). Five of these TFs showed slight but significant expression differences between the fins (Fig. 4A), however only one TF, *irf1*, showed dorso-ventral differences (D, Cd < A, Cv). This pattern was confirmed in comparisons among anatomically equivalent regions, as the reduced expression of *irf1* in the black stripe regions (first row in Fig. 4B) was robust against the correction for melanophore density (last row in Fig. 4B).

We also tested for expression correlations among the seven differentially expressed target genes (identified in Fig. 3A) and the six predicted TFs (Fig. 4C). The results showed positive expression correlations (blue shadings in Fig. 4C) between several pairs of target genes, including *igsf11* and *pmel*, as well as negative expression correlations (red shadings) between target genes and three TFs, *irf1*, *irf2* and *ap2a*. Notably, strong expression correlations with several target genes were observed for *irf1*, which was negatively correlated with *igsf11*, *mitfa*, *pmel* and *slc45a2*.

## DISCUSSION

Patterns of dark and light stripes are found in many fish species and may not only reflect spatial variation in the number of chromatophores, but can also arise from differences in the pigmentation of the individual chromatophores (*Parichy, 2006*; *Greenwood, Cech & Peichel, 2012*). In the here investigated cichlid fish, *Neolamprologus meeli*, the dorso-ventrally distinct fin stripes involved differences in the spatial distribution of a constant number of melanophores, i.e., their aggregation and depletion in the dorsal black and white stripes, respectively, versus a more homogeneous melanophore density in the ventral fin regions. Although melanophore density in the *N. meeli* fin samples showed the expected gradient between black, grey and white fin regions, its correlation with the expression level of the melanosome marker *slc24a5* was not consistent across fin regions. *Slc24a5* encodes an intracellular membrane cation exchanger predominantly present in melanosomes. Its tissue expression level has been shown to be associated with the number of melanin-producing cells, and reduced expression is associated with fewer, smaller and less pigmented melanosomes (*Lamason et al., 2005*). The mechanism by which *slc24a5* expression can be regulated differently across melanophore subpopulations in distinct body parts is unclear but it might involve the nonsense-mediated mRNA decay pathway (*Wittkopp et al., 2009*). Our data suggest that the melanin-based colour differences among fin regions of *N. meeli* are primarily due to differences in melanophore density, with additional variation contributed by variation in melanophore pigmentation.

The light-reflecting iridophores are not individually discernible, and we used the expression of the iridophore marker gene *ltk* to confirm their presence and trace their distribution in the fin tissue. *ltk* encodes a member of tyrosine kinase receptors and its expression is essential for fate specification of iridophores and formation of iridophore-containing stripes in adult zebrafish (*Lopes et al., 2008*). *Ltk* expression was detected in

all fin regions, but was significantly elevated in the white stripes, which is consistent with an aggregation of iridophores in these regions. This iridophore aggregation could also be involved in the development of the black stripe distal to the white stripe. In the zebrafish dermis, iridophores repel melanophores on the short range but support their aggregation at a longer distance, and thus contribute to melanophore stripe formation (*Frohnhöfer et al., 2013*).

Since the stripe pattern of the *N. meeli* fins is dorsally restricted, we were primarily interested in genes displaying a corresponding dorso-ventral difference in their expression levels. Two of the investigated candidate genes, *igsf11* and *pmel*, followed this pattern in comparisons among fins as well as among analogous fin regions after correction for melanophore density. Expression levels of both genes were higher in the dorsal black and white stripe regions than in the corresponding ventral fin regions (Fig. 3B). *Igsf11* encodes the immunoglobulin superfamily member 11, a classical cell adhesion molecule, which mediates cell adhesion and promotes the migration and survival of melanophores and their precursors during adult stripe formation in zebrafish (*Eom et al., 2012*). Interestingly, mutations in *igsf11* can have local effects. While a homozygous mutation in the fourth exon of *igsf11* induces severe irregularities in both body and fin stripes of adult zebrafish, defects caused by mutations in the second exon are mainly restricted to ventral body stripes but not apparent in fins (*Eom et al., 2012*). Since the melanophores of zebrafish *igsf11* mutants failed to migrate, the higher expression of *igsf11* in both black and white stripes of *N. meeli* might suggest an increased motility of melanophores in these regions. Furthermore, increased *igsf11* expression may also affect the arrangement of iridophores. Although the function of *igsf11* in iridophores is unclear, Tjp1a, a tight junction protein recently identified as regulator of iridophore organization in zebrafish stripes, has been suggested to be an interacting partner of igsf11 protein in relevant cellular processes (*Fadeev et al., 2015*).

The other gene with dorsally elevated expression levels, *pmel*, encodes a melanosome protein (Silver), which plays an essential role in the structural organization of premelanosomes and the formation of intra-lumenal fibrils during melanosome biogenesis (*Schonthaler et al., 2005*). In homozygous zebrafish mutants of *pmel*, changes in the shape positioning and melanin content of melanophores result in hypopigmented adults (*Schonthaler et al., 2005*). The function of *pmel* in the formation of pigment patterns has not been investigated, however, its differential expression was suggested as a contributing factor to the dark-light stripe formation in freshwater threespine sticklebacks compared to marine sticklebacks with evenly distributed melanophores and iridophores throughout skin (*Greenwood, Cech & Peichel, 2012*).

The increased expression of *igsf11* and *pmel* in both black and white regions of the fins might represent an overlapping transcriptional signature of melanophore and iridophore subpopulations emanating from shared developmental origins in the neural crest (*Curran et al., 2010*). In zebrafish, an attempt to find overlapping transcriptional signature between melanophores and iridophores identified 62 genes with similar expression patterns in both chromatophores, but *igsf11* and *pmel* were not among them (*Higdon, Mitra & Johnson, 2013*). However, the study was done using larval melanophores and iridophores from the body and not from the fins (*Higdon, Mitra & Johnson, 2013*). This could imply that the

dorsally distinct stripe pattern in *N. meeli* fins involves a novel regulatory mechanism with locally restricted activity in the dorsal fin regions, which induces *igsf11* and *pmel* in both chromatophore classes and affects cell migration and melanosome biogenesis.

We also found an overall positive expression correlation between *igsf11* and *pmel* across all fin regions suggesting a potential co-regulation through a shared upstream transcriptional regulator. There is no evidence yet for such a mechanism in fish, but a recent computational study in human has predicted TFs, including Ap2a and Mitf, and small regulatory RNAs, such as mir-221, as shared upstream regulators of *igsf11* and *pmel* in specific melanocyte lineages (*Rambow et al., 2015*). Here, we have already tested *mitfa* expression in our candidate gene survey and did not detect elevated expression in the dorsal black and white stripes In addition, *mitfa* showed positive expression correlation with *pmel* but not with *igsf11* (Fig. 4C). Among the shared TFs predicted by the motif enrichment approach employed in the present study, only *irf1* displayed dorso-ventrally distinct expression (with higher expression level in ventral regions) (Figs. 4A and 4B). Moreover, *irf1* showed negative expression correlations not only with *igsf11* and *pmel* but also with *mitfa* and *slc45a2* across the fin regions (Fig. 4C). The gene encodes a member of the interferon regulatory transcription factor family and mediates a cytokine-dependent hypopigmentation process in human and mice melanocytes (*Natarajan et al., 2014*). The mechanism, by which Irf1 impedes pigmentation, appears to be involved in melanosome maturation and is independent of Mitf regulation (*Natarajan et al., 2014*). The expression pattern of *irf1* in the *N. meeli* fins, the enrichment of its binding sites in *igsf11* and *pmel* promoter regions together with the regulatory role of both *irf1* and *pmel* in melanosome biogenesis suggest *irf1* as a strong candidate for the regulation of the observed fin colour phenotype. More specifically, irf1 might act as suppressor of both *igsf11* and *pmel* in the chromatophores.

Finally, we note that the involvement in dorso-ventrally distinct colour patterning of those genes, which did not show the expected expression in our study, should not be discarded based solely on mRNA expression data. For instance, our TF prediction also included *ap2a*, which encodes a TF essential for subsets of neural crest derivatives, including subpopulations of pigment cells. Its loss-of-function mutation in zebrafish leads to defects in the migration and differentiation of melanophores and iridophores, but not xanthophores (*Knight et al., 2004*; *Li & Cornell, 2007*). Interestingly, the zebrafish *ap2a* mutant exhibits a reduced number of iridophores and melanophores in ventral and lateral stripes but not in dorsal stripes (*Knight et al., 2004*). Similarly, locally restricted effects on stripe patterning are known for a structural mutation in *igsf11* (*Eom et al., 2012*). Therefore, while our results mark *pmel*, *igsf11* and *irf1* as likely components of the distinct dorso-ventral fin patterning in *N. meeli*, additional studies are needed to clarify the roles of other candidate genes. It is also important to investigate whether the same genes are involved in stripe pattern formation during development, and particularly, at early emergence of the pattern in fins.

## CONCLUSIONS

Variation in colours and patterns contributes significantly to the tremendous phenotypic diversity among fishes. Elucidating the molecular basis of this diversity is a stimulating challenge to research in both model and non-model species, and promises significant insight in the mechanisms that translate sexual and natural selection pressures into phenotypic variation. In our work, we capitalized on a naturally occurring, dorsally restricted fin pattern phenotype to investigate the mechanisms behind colour pattern differentiation in dorsal versus ventral regions. The investigated fin colour pattern was associated with variation in melanophore and iridophore densities, and corresponded with the gene expression patterns of two candidate target genes, *igsf11* and *pmel*, and their predicted shared upstream regulator *irf1*. Further studies are required to identify functional relationships between these genes and other potential components of this process.

## ACKNOWLEDGEMENTS

The authors thank Wolfgang Gessl for his responsible management of our fish facility and photography of the fishes. We also thank Holger Zimmermann and Stephan Koblmüller for sharing their precious knowledge on cichlid fishes of Lake Tanganyika, as well as Silke Werth and the Institute of Plant Sciences at University of Graz for technical assistance.

### Funding

The University of Graz provided financial support for this study. The funders had no role in study design, data collection and analysis, decision to publish, or preparation of the manuscript.

### Grant Disclosures

The following grant information was disclosed by the authors:
University of Graz.

### Competing Interests

The authors declare there are no competing interests.

### Author Contributions

- Ehsan Pashay Ahi conceived and designed the experiments, performed the experiments, analyzed the data, contributed reagents/materials/analysis tools, wrote the paper, prepared figures and/or tables, reviewed drafts of the paper.
- Kristina M. Sefc conceived and designed the experiments, analyzed the data, contributed reagents/materials/analysis tools, wrote the paper, prepared figures and/or tables, reviewed drafts of the paper.

### Animal Ethics

The following information was supplied relating to ethical approvals (i.e., approving body and any reference numbers):

The biopsies were performed under permit number BMWFW-66.007/0013-WF/V/3b/2016 issued by the Federal Ministry of Science, Research and Economy of Austria.

## Data Availability

The raw data has been supplied as Supplementary File.

## Supplemental Information

Supplemental information for this article can be found online at http://dx.doi.org/10.7717/peerj.2843#supplemental-information.

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
