# Peer review of "A gene expression study of dorso-ventrally restricted pigment pattern in adult fins of Neolamprologus meeli, an African cichlid species"

_PeerJ, doi:10.7717/peerj.2843_

## Round 0.1 · original submission · Minor Revisions

Three reviewers have provided detailed feedback. Please consider all their suggestions in the revised paper.

Reviewer 1 ·

Basic reporting

The article is easy to follow and the experiment sufficiently explained. I miss a bit more introduction to the fish system that has been used, especially to the development of the color pattern. Is this fin patterning characteristic for adult fish and how is it established? How much plasticity do you observe among individuals or in other words, are fins constantly pigmented in the described way? Do you expect a difference in gene expression throughout development, in other words, are different genes involved in establishing the phenotype in the first place than in maintaining it in the adults? This was not always clear to me in the discussion.

Experimental design

The study design is well explained. Following up on my comment above, I am only wondering if the sample size of 4 is adequate. It should probably be increased if there is variation in the color pattern. To my knowledge, analyses performed are in agreement with the standards in the field.
The weakest point of the experimental set-up is probably the usage of another species (O. niloticus) for the promoter sequence analysis given the fact that also only 1kb of sequence was analyzed. If this sequence represents the core promoter region and conservation between cichlid genomes is assumed for it, a simple PCR amplification from N. meeli followed by amplicon sequencing would probably have given the most reliable insight and should be relatively easy to carry out.

Validity of the findings

All data are provided and all tests explained. Results from statistical tests should be submitted as supplementary files (e.g. anova plus post-hoc tests).
The authors are aware of the limited power that a targeted gene candidate approach has and hence formulate conclusions with appropriate caution.

Additional comments

Overall, I enjoyed reading this article and the way data are presented is well done.
It would be nice to have a real fish image, accompanying figure 1 for illustrative purposes.
The results section would benefit from subheadings and a short introduction to the chosen candidate genes. On which criteria were they selected?
In figure 1B/C please add the gene/mutation below the phenotype or explain better in the legend.

·

Basic reporting

In this study, the authors aim to help elucidate the mechanisms behind fin stripe patterns in cichlid fish. Using Neolamprologus meeli as a model, they examine the expression levels of candidate color genes in fins displaying different arrangements of pigmented cells into stripes. They identifiy two genes, pmel and igsf11, with differential expression along the dorso-ventral axis. To investigate further, they identified potential transcription factors associated with the regulation of these genes, and found that irf1 is negatively correlated with their expression. They conclude that pmel, igsf11, and irf1 may be involved in dorso-ventral fin color patterning in this species. I recommend to accept the article with minor revisions.

Experimental design

Overall, the study addresses a relevant, current topic in Developmental/Evolutionary Biology using an exciting model system and an appropriate experimental design. The methods and analysis are adequate. I have only minor concerns:

- It would be helpful to see a picture of the whole fish (instead of a drawing) and high resolution images of all analyzed fins (at cellular resolution to be able to see chromatophores)

Validity of the findings

The authors draw reasonable conclusions without over interpreting their data. However, there are a few minor concerns:

- One caveat of the study is the fact that gene expression is only examined in adult fish. The authors could emphasize that more clearly. The distinction is made in Table 1, but it should be stated more explicitly in the main text. Obviously, gene expression changes during development might be causal to coloration phenotypes, and those might not necessarily be maintained in adult fins. Analyzing gene expression pattern in development would be beyond the scope of this paper, still I believe it is an important point to be discussed.
- The authors do not account for the presence of xanthophores in their samples, although several genes tested are known to be involved also in xathophore development/maintenance (as stated in Table 1). Maybe the authors could expand on that (i.e. also by showing stainings for Xantophores; see i.e. PMC4352272) or state more clearly why they ignored xanthorphores.
- Figure 1B/C Ednrb2 is not analyzed in this study. I feel that this section of the figure could be omitted. While they help to put the study in a broader context, after reading the complete manuscript the link to melanocyte pattern phenotypes in mammals and birds feels overemphasized, compared to their relevance to the study.
- A graphical summary summarizing the expression patterns of pmel1 and igsf11 and a model how they act and are controlled differently by irf1 in the different fins and fin regions would definitely help the reader to grasp the results of this study more easily.

Additional comments

Line 225+: P-values for differences in melanophore density should be included in the main text
Line 398-400: The authors briefly mention the function of irf1, but a more detailed description would help the reader
Lines 64-66: "Studies in ...": Sentence is a little unclear
Line 122-125: The abbreviations of each find region are shown clearly in figure 1A, but the description of this system in text and the other figures is slightly confusing.
Line 241:-246: This information would probably fit better in the methods section.
Table 1: "Xentaphore"

Reviewer 3 ·

Basic reporting

Basic Reporting:

Overall the introduction, background and results are presented in a clear, logical manner. However, I have some suggestions to improve the overall clarity of the work.

1. Use of the words “differentiate” and “differentiation”
Throughout the paper, the words differentiate and differentiation are used to describe differences between gene expression or pigment patterns on the dorsal and ventral sides of the fish. While in a strict grammatical sense, this usage in correct, it is confusing to a Biological audience because “differentiate” has a very specific definition in Developmental Biology. I would therefore recommend changes “differentiation” to “differences” throughout the text.

2. Use of the word dorsalized to describe the pattern
Similarly, the word “dorsalized” is used in Developmental Biology to describe an embryonic phenotype in which the embryo forms only dorsal tissues. Again, its usage here is confusing to a biological audience and I would instead suggest referring to the patterns as “dorsal stripes” or stripes localized to the dorsal side (dorsum).

3. Simplification of data presentation
While great effort was made to connect the pattern (color) to the region of fin being examined, the overall effect was to make the graphs more confusing rather than less. My suggestion would be to number the regions within each fin (1=distal, 2=intermediate, 3=proximal). The reader can then go back and reference Figure 1 and Figure 2 to remind themselves which regions are black, white or gray. I would also suggest using a capital letter to define each fin (D=dorsal, C=caudal, A=anal), followed by a lowercase letter to distinguish between the dorsal and ventral halves of the caudal fin (Cd and Cv rather than dc and vc). I am attaching a pdf with my suggestion for a revised labeling scheme below.

4. Tukey-HSD post hoc significance
Currently the bar graphs show only the groups that ARE significantly different above the bars. In part because the labels are so long, this is somewhat confusing. Additionally, it seems somewhat arbitrary which bars are labeled. I would instead suggest designating all the groups that are NOT significantly different with a letter (ex. a, b, c, etc.). Groups that are significantly different would be labeled with a different letter. In this manner, every bar on the graph would have a letter (in some cases two) above it and it would be easy to quickly see which groups are significantly different from one another (they would be labeled with different letters) and which are not for all the possible comparisons.


In the text of the manuscript, I also have some specific line by line suggestions to improve the overall clarity of presentation:

Line 15: change “belong to” to “are one of the”
Line 16: delete “the variation in”
Line 21: change “dorsalized” to something else, like “a distinct dorsal pattern”
Line 23-24: change “differentiation” to “differences in dorsal and ventral patterning”
Line 29-30: again “differences in the”
Line 34: “differentiation” needs to be changed, suggest: “revealed a difference in expression on the dorsal and ventral sides”
Line 36: “differentiation” should be changed. Suggest “controlling differences in fin pattern development between the dorsal and ventral sides of the body”.
Line 42: Replace “owned” with due
Line 54: Add leucophores, white cells in medaka with appropriate reference
Line 59: delete “degrees of”
Line 65: delete “rather tends to” replace with “instead”
Line 69: Organization might be a better word choice than distribution
Line 83: change “dorsalized”. Suggest: A distinct pattern of light and dark stripes on the dorsal side and a lack of stripes on the ventral side.” Or dorsum and ventrum.
Line 84: “differentiation” should be changed
Line 88: “differentiation” should be changed
Line 90: not just density, it could also be the number of melanosomes
Line 91: Proliferation should be replaced with number or abundance. Cell proliferation is not assessed in this manuscript.
Line 106: “dorsalized” should be changed. Suggest: dorsal stripes.
Line 110: I would appreciate the addition of a sentence here stating that males and females have the same pattern in their fins as this is not the case for all species of fish.
Line 120-125: If you take my suggestion for changing the naming scheme, this section will need to be re-written.
Line 237: Absence not depletion
Line 278: Should this be number instead of density?
Line 280: “differentiated” should be changed
Line 281: “Dorso-vental differentiation” should be changed. Also be careful when referring to the dorso-vental axis. Axis seems to suggest the entire body from the dorsal side to the ventral side when really the dorsum (dorsal side) is being compared to the ventrum (ventral side). Expression along the entire axis is not being assessed.
Line 284-285: Something about this sentence doesn’t seem quite right.
Line 286: delete “levels”
Line 293: Shared neural crest origin? Lots of cells have shared neural crest origin. If this is a specific statement about shared origin of melanophores and iridophores then there should be a reference here (Curran et al, 2010, Developmental Biology 344(1): 107-118).
Line 311: Last Row in Figure 4B not 4A
Line 310: First row in Figure 4B not 4A
Line 323-328: This needs to be rewritten and broken into at least two sentences. And again, “dorso-ventral differentiation” needs to be changed to differences in dorsal and ventral sides or patterns.
Line 350: “dorsalized” should be changed. Try “restricted to the dorsal side” or “localized to the dorsal side”. Maybe “identify genes that also show a similar difference in expression between the dorsal and ventral fins” Or “with expression that is higher on one side than the other”
Line 380: Cite reference, see above for line 293.
Line 383: This work was done using larval melanophores and iridophores from the body and not from the fins. Both the difference in developmental timing and location might explain why these cells showed different expression than adult fins.
Line 396: Again, not all along the axis. The dorsal side compared to the ventral side. Also, clarifying in the text which side showed higher levels of expression would be helpful.
Line 403-404: So is the model that ir1 expression in the ventral fins is suppressing igsf11 and pmel expression? If so, make more explicit.
Line 412: Ap2a mutants have defects in melanophore migration. Since melanophores originate at the dorsal side of the neural tube, a mutant with defects in melanophore migration will likely show a more severe phenotype on the ventral side than the dorsal side because melanophores have further to migrate to reach the ventral side. This difference is due to migration and not dorsal vs ventral patterning.
Line 426: “Dorsalized” should be changed
Line 427: “Dorsal side vs ventral side” not along the axis.

For the figures I would like to make the following suggestions:
I would also recommend changing the title of Figure 1 to remove “Dorsalized”. Perhaps replacing it with dorsally restricted melanization or stripes.

For figure 2, part C, I would suggest using the same scale for all graphs.

For the Figure Legends, I would add that the comparisons between groups were done specifically by using Tukey-HSD.

Experimental design

The overall goal of this manuscript seems very clear: to identify gene expression differences that underlie a variation in pigment patterning in the dorsal fins compared to the ventral fins of N. meeli.
The quantitative and technical aspects of these analyses were carried out in a rigorous manner and really set the standard for future gene expression studies. The overall methods are well described and will be useful to other researchers trying to identify the best control genes for their qPCR experiments. Identification of potential transcription factor binding sites in the promoter regions was also well described, but it would be helpful to clarify whether this analysis was done for only upstream promoter regions or whether intronic regions where also analyzed. In zebrafish some of the large early introns have been shown to have important regulatory functions.

Validity of the findings

Overall the data seem robust and statistically sound. The statistical tests used are appropriate for the type of data being analyzed. Trying to correct the gene expression levels for the number of melanophores/mm2 is an unexpected way to look at the data, but seems like a reasonable way to identify genes that are truly being expressed at higher levels in melanophores compared to expression that is higher simply because there are more melanophores in that area. The best way to look at this would be to do RNAseq comparisons between melanophores in the dorsal fin compared to melanophores in the ventral fin, but this a suitable first step.

Overall the conclusions are appropriately stated, although they could be clarified. Overall there are two major conclusions from this manuscript. 1) igsfll and pmel are expressed at higher levels in the dorsal fins – fins with distinct dark and light stripes and 2) that ir1 is expressed at higher levels in the ventral fins. Because there are potential ir1 binding sites in the promoter sequences of igsf11 and pmel, these results suggest that igsf11 and pmel are being suppressed in the ventral fins by the higher expression of ir1.

Of course, this raises the question of what controls ir1 expression?

Some of the speculation in the discussion section seems to confuse effects due to differences in migration compared to patterning mechanisms. Genes that are expressed cell autonomously (most of the genes analyzed in this manuscript) and effect migration can result in defective patterns even if the patterning mechanisms are intact. This would be the case for the ap2a mutant. Likewise, discussion of igsf11 should take into account that igsf11 has been shown to be expressed by melanophores and is involved in both migration and adhesion. This makes speculation about its role in patterning. Additionally, igsf11 has been shown to be expressed by a variety of tissues in addition to pigment cells. Since this manuscript does not include any in situ data, conclusion about which cells are expressing igsf11 and depend upon its expression, should be tempered.

I think a little refinement of the discussion section would improve the manuscript but the overall conclusion follows logically from the experiments.

Additional comments

The experimental and technical aspects of this paper were excellent. My major concerns with the manuscript are the use of the words "differentiate", "dorsalized" and "dorso-vental axis" which have very specific meanings in Developmental Biology that are very different from the way they are being used here. The manuscript will be strengthened by changing these word choices.

Annotated reviews are not available for download in order to protect the identity of reviewers who chose to remain anonymous.

---

## Round 0.2 · accepted · Accept

Thank you for improving your manuscript.

Reviewer 1 ·

Basic reporting

No Comments

Experimental design

No Comments

Validity of the findings

No Comments

Additional comments

The authors have done a god job in integrating the reviewers comments and improved the manuscript, I have no further suggestions on the revised version.

·

Basic reporting

The authors have addressed all my comments. I would like to congratulate the authors to their work and their thorough revision and recommend publication.

Experimental design

-

Validity of the findings

-

Additional comments

-

Reviewer 3 ·

Basic reporting

This manuscript meets the PeerJ standards of basic reporting.

Experimental design

This manuscript meets the PeerJ standards of experimental design.

Validity of the findings

This manuscript meets PeerJ standards of validity of findings.

Additional comments

Overall, I felt the new version of the manuscript was improved and appreciated the authors careful consideration of reviewers' suggestions. I support publication as this manuscript.